# Genetic variation of the mitochondrial DNA control region across plains bison herds in USA and Canada

**Gaimi Davies**[1¤], **Blake McCann**[2], **Lee Jones**[3], **Stefano Liccioli**[4], **Maria Cecilia Penedo**[5], **Igor V. Ovchinnikov**[1]*

**1** Department of Biology, University of North Dakota, Grand Forks, North Dakota, United States of America, **2** Theodore Roosevelt National Park, Medora, North Dakota, United States of America, **3** United States Fish and Wildlife Service, Natural Resource Program Center – Wildlife Health, Bozeman, Montana, United States of America, **4** Grasslands National Park, Parks Canada Agency, Val Marie, Saskatchewan, Canada, **5** Veterinary Genetics Laboratory, University of California, Davis, California, United States of America

¤ Current address: Turner Institute of EcoAgriculture, McGinley Ranch, Gordon, Nebraska, United States of America

* igor.ovtchinnikov@und.edu

**Data Availability Statement:** All relevant data are within the paper and its Supporting information files.

## Abstract

Once numbering in the tens of millions, bison (*Bison bison*) have faced a broad range of challenges over the past century, including genetic impacts from the population bottleneck in the 1800s, and significant loss and fragmentation of habitat resulting in little opportunity for population growth on remaining small, geographically isolated reserves. To identify best practices for bison stewardship against this backdrop, managers must understand the genetic composition of existing conservation herds. This study characterized 14 plains bison (*Bison bison bison*) herds managed by the U.S. Department of Interior and Parks Canada Agency based on complete mtDNA control region sequences. Among 209 bison, we detected 11 major mtDNA control region haplotypes based on nucleotide substitutions and 23 sub-haplotypes where indels are considered. We determined matrilineal relationships between the herds and compared our genetic findings to historic records. The recent common ancestry of modern bison deriving from small, scattered groups combined with gene flow through foundation and translocation events between herds during the last 100 years, is reflected in Fst value (0.21), haplotype (0.48 ± 0.04) and nucleotide (0.004 ± 0.002) diversities, and mean number of pairwise differences (3.38 ± 1.74). Genetic diversity was distributed unevenly among herds, with 21.41% of genetic variation observed between herds. Median joining network, together with trends in the Tajima's D and Fs tests, revealed two patterns in the recent evolution of mtDNA sequences in bison: mutational process has generated diversity with a Hap 1 haplotype epicenter, and missed mtDNA haplotypes exist in the network due to bottleneck, loss through management practices, or incomplete sampling of specimens across conservation herds. This work significantly expands characterization of the genetic diversity among bison conservation herds, providing additional

**Funding:** This study was partly supported by the National Park Service (www.nps.gov) agreement (P16AC00001) to I.V.O. and B.M. The funder had no role in study design, data collection and analysis, decision to publish, or preparation of the manuscript.

**Competing interests:** The authors have declared that no competing interests exist.

decision support for managers considering restoring gene flow to achieve long-term species viability.

## Introduction

Bison (*Bison bison)* occupied the North American landscape, from Alaska to Florida and from the Rocky Mountains to the Appalachian Mountains, in the tens of millions before their near extirpation in the late XIX century [1]. Only small, dispersed remnants of the two North American subspecies, the plains bison (*Bison bison bison*) and the wood bison (*Bison bison athabascae*) amounting to less than 1000 individuals, were preserved [2, 3]. Early conservationists recognized the consequences of widespread population loss and began efforts to recover bison, gathering small groups of remaining animals from which all modern herds descend [4]. Bison restoration efforts continue today but face a broad range of challenges.

At present, ~30,000 plains and wood bison are kept in geographically isolated conservation herds [5]. The majority of these herds are relatively small (i.e., < 1000 individuals), with little opportunity to grow due to conflicting land use associated with cattle grazing and conversion to agriculture [6, 7]. Additionally, the population bottleneck in recent bison history leaves the species susceptible to a global loss of genetic variation through genetic drift, founder effect, and inbreeding, which represent further threats to their long-term persistence. While some authors recommend managing bison as a metapopulation in these fragmented habitats and complicated socio-economic mosaic [8, 9], the genetic composition of existing conservation herds must be first assessed and taken into account.

Historical records suggest that the origin and genetic composition of most modern bison herds may be complex due to sourcing from multiple locations, augmentation from a variety of herds of unknown or mixed origin, and historic cross breeding with domestic cattle. While some studies have undertaken an assessment of genetic variability in conservation bison herds using microsatellite analysis [9], few have used the control region or complete mitochondrial DNA (mtDNA) sequences to infer the diversity of maternal lineages, their variability and origin.

The first estimate of bison mtDNA variation was carried out in five U.S. (Custer State Park, Fort Niobrara Wildlife Refuge, National Bison Range, Wichita Mountains Wildlife Refuge, Yellowstone National Park) and three Canadian (Elk Island National Park, Mackenzie Bison Sanctuary, Wood Buffalo National Park) conservation herds. Restriction digestion of mtDNA control region, a small non-coding section of the mitochondrial genome, was utilized to detect variable sites in 269 bison followed by DNA sequencing of a segment of the control region in a small group of 32 bison [10]. Although the authors sought to understand the history of differentiation between wood and plains bison subspecies, the phylogeny of eleven bison mtDNA haplotypes demonstrated that neither wood bison nor plains bison represent a well-defined taxon [10].

DNA sequencing of a 677-bp section of the mtDNA control region was also conducted in eleven U.S. (Antelope Island State Park, Custer State Park, Finney Game Refuge, Fort Niobrara Wildlife Refuge, Henry Mountains, Maxwell Game Refuge, National Bison Range, Wichita Mountains Wildlife Refuge, Williams Ranch, Wind Cave National Park, Yellowstone National Park) and three Canadian (Elk Island National Park, Mackenzie Bison Sanctuary, Wood Buffalo National Park) herds. Eight unique mtDNA haplotypes were identified in 53 bison not carrying domestic cattle mtDNA. These eight haplotypes were unequally distributed amongst the sampled herds from U.S. and Canada, with the most common haplotype found in 25 of 53 animals across eight herds, and no haplotypes were identified as unique to any herd [11].

Douglas et al., 2011 [12] expanded upon previous work by sequencing the entire mitochondrial genome, including samples from 29 plains bison and 2 wood bison from Elk Island National Park, Fort Niobrara National Wildlife Refuge, National Bison Range, Texas State Bison Herd, Yellowstone National Park, and a private herd from Montana. They found that sixteen whole mitochondrial DNA sequences generated two clades in a maximum likelihood phylogeny. Clade I consisted of plains bison from Fort Niobrara National Wildlife Refuge, Yellowstone National Park, the National Bison Range, and a private herd. Clade II was comprised of plains bison from the Texas State Bison Herd as well as 2 wood bison from Elk Island National Park [12].

While previous authors characterized the bison mtDNA variation in a variety of public and private bison herds of the U.S. and Canada, several conservation herds were excluded from these studies. Furthermore, even the largest study of mtDNA variation in conservation herds was completed over 20 years ago. To date, there are no comprehensive studies of the origin and evolution of mtDNA across the majority of U.S. Department of Interior (DOI) and Parks Canada Agency (PCA) plains bison herds that include comparison of historic and genetic records. With this work, we: 1) characterize the mtDNA control region variation of 14 DOI and PCA herds; 2) determine phylogenetic relationships among herds; and 3) provide recommendations for herd conservation management based on our analysis.

## Materials and methods

### Collection of bison specimens

Bison samples were collected via biopsy dart under NPS IACUC "MWR.IMR.AKR_THRO. CHIC.GRCA.WRST_Bison_2016.A3" or through processing of hair and blood samples collected ancillary to bison capture and handling operations as part of routine management across herds from 2015–2017. Research was conducted under "NPS 21st Century Bison Conservation Management" project and Parks Canada research and collection permit number EI-2018-28978. We included samples from 14 plains bison herds established between 1907 and 2009. Five of these populations are managed by the US Fish and Wildlife Service (FWS): Fort Niobrara National Wildlife Refuge, National Bison Range, Neal Smith National Wildlife Refuge, Rocky Mountain Arsenal National Wildlife Refuge and Wichita Mountains Wildlife Refuge. Five populations are managed by the US National Park Service (NPS): Badlands National Park, Tallgrass Prairie National Park, Theodore Roosevelt National Park, Wind Cave National Park, Wrangell—St. Elias National Park. Theodore Roosevelt National Park manages two geographically separate herds, but these have been combined in our analysis due to common origin. The herd migrating between Grand Teton National Park and National Elk Refuge is co-managed by both the NPS and FWS. The Book Cliffs herd is managed cooperatively by the State of Utah and the Ute Tribe, occurring on lands administered by the Bureau of Land Management. Finally, two herds in our study (Elk Island National Park and Grasslands National Park) are under the Parks Canada Agency's jurisdiction.

### DNA isolation, PCR, and mtDNA sequencing

We sequenced the mtDNA control region for samples collected from 15 randomly selected animals from each herd, with the exception of Wind Cave National Park where n = 14. Based on results of prior screening for cattle introgression markers [9], all the selected samples had bison-origin mtDNA, with no evidence of cattle introgression.

DNA was extracted from 5–6 hair roots or from 3 mm diameter punches of skin tissue with a standard Proteinase-K digestion protocols used by the Veterinary Genetics Laboratory, U.C. Davis [13]. We used two PCR primer sets (Bb-16163For: AAACCAGCAACCCGCTAG/ Bb-

413Rev: `GACTCATCTAGGCATTTTCAGTG` and Bb-15311For: `CCCCACATCAAACCCGAATG`/
Bb16271Rev: `GCCCTGAAGAAAGAACCAGATG`) to amplify two overlapping fragments spanning the bison mtDNA control (D-loop) region, each with approximately 573 bp (Amplicon 1) and 960 bp (Amplicon 2), respectively. PCR reactions prepared in 25 μl total volume contained 0.2 μM of each primer pair, 1x Reaction Buffer IV (ThermoFisher Scientific), 2 mM MgSO₄, 1.5 units of Go Taq G-2 Flexi (Promega) and 3 μl of DNA template. PCR cycling consisted of 35 cycles at 94˚C for 30 seconds, 55˚C for 45 seconds and 72˚C for 90 minutes, with a final extension at 72˚C for 10 minutes.

Direct cycle sequencing was performed with the BigDye Terminator v3.1 kit (Applied Biosystems), with the forward and reverse primers for each fragment, and internal forward sequencing primers Bb-1Seq2-F: `ATGCTTGGACTCAGCTAT` for Amplicon 1 fragments, and Bb-2Seq2-F: `TGAAGACAGGTCTTTGTAGTACA` for Ampicon 2 fragments. Products of the sequencing reactions were separated on AB 3730 DNA Analyzer (ThermoFisher Scientific) instrument and electropherograms were analyzed with SeqMan Pro™ v10.0.1 (DNASTAR Inc.). For each sample tested, all six trace files were assembled and analyzed to produce a contig of approximately 1333 bp. For 22 samples, where sequence quality of the Amplicon 1 (~573 bp) fragment was lower, amplicons were also cycle-sequenced with internal primers Bb-1Seq3-F: `CAGAGGATCCCTCTTCTCG` and Bb-1Seq4-R: `GGGCCTGCGTTTATATATTG`.

## Analysis of bison mtDNA control region variation

We selected GenBank accession number GU946990, from Montana, as a reference in our work. The mtDNA sequences were aligned to the reference using Mega7 software [14]. For comparative analysis, mtDNA contigs were trimmed to the control region corresponding to nucleotide positions 15,794–16,323 and 1–362 of the reference.

Variabilities within and amongst bison populations were estimated using haplotype and nucleotide diversity in DnaSP version 6 [15]. Because the fixation rate for small insertions and deletions (indels) in bison mtDNA is unknown, these were excluded from population statistical analysis. Genetic differentiation such as mean pairwise differences, $F_{ST}$, and Analysis of Molecular Variance (AMOVA), Tajima's D, raggedness index, and Fu' values were calculated using ARLEQUIN version 3.5.2.2 [16]. AMOVA was performed for all bison groups using pairwise differences with 1023 permutations. The distance matrix of pairwise $F_{ST}$ values between bison populations was calculated using pairwise differences and 100 permutations to obtain p-values.

For alternate comparison of genetic signal between bison populations, we used metric Multi-Dimensional Scaling (MDS) in R Studio version 1.1463 (http://www.rstudio.com/). Genetic distances used for MDS were based on pairwise $F_{ST}$ values between populations. The "ggplot2" R package [17] was used to modify graphs.

A median joining network tree was constructed to show relationships between haplotypes using Population Analysis with Reticulate Trees (PopART) version 1.7 [18]. Epsilon was set to zero with 500 permutations to draw the network.

## Results

From a total sample of 209 bison from 14 North American plains bison herds, we found 23 haplotypes determined by 15 transitions, 4 insertions and 5 deletions of single nucleotides (collectively referred to as indels) in the control region, determined by nucleotide positions of 15,794 and 362 in the reference sequence (S1 Table). MtDNA haplotypes were deposited at NCBI GenBank with accession numbers OM791248 –OM791270. Because the fixation rate for indels is unknown and the same indels can independently emerge in different mtDNA

**Table 1. MtDNA control region haplotypes determined by single nucleotide polymorphisms (SNPs) found in 14 U.S. Department of Interior and Parks Canada Agency bison herds.**

| Haplotype | SNPs (versus the reference GU946990) |
|---|---|
| Hap 1 | none |
| Hap 2 | 16248 G→A |
| Hap 3 | 15895 C→T, 16122 C→T, 16189 T→C, 16283 A→G, 166 A→G |
| Hap 4 | 15895 C→T, 16040 C→T, 16050 C→T, 16122 C→T, 16131 T→C, 16283 A→G |
| Hap 5 | 15895 C→T, 16122 C→T, 16283 A→G, 166 A→G |
| Hap 6 | 15895 C→T, 16122 C→T, 8 G→A |
| Hap 7 | 15895 C→T, 15957 A→G, 15965 C→T, 16042 T→C, 16122 C→T, 16283 A→G |
| Hap 8 | 16041 C→T |
| Hap 9 | 16283 A→G |
| Hap 10 | 16122 C→T |
| Hap 11 | 15895 C→T, 16122 C→T, 16279 C→T |

Each single nucleotide polymorphism (SNP) indicated in position corresponding to the GU946990 mtDNA sequence. Arrow (→) shows the replacement of the reference nucleotide by the derived nucleotide found in this study.

lineages, only 11 major haplotypes (Hap 1 –Hap 11) defined by 15 transitions and without indels (Table 1) were included for the analysis of the haplotype composition of bison herds.

The 11 major haplotypes found in bison herds were unevenly distributed among the DOI and PCA herds (Fig 1; S2 Table). The most common Hap 1 was found in 148 (71%) bison and in all 14 herds (S2 Table), and included several sub-haplotypes differing in 4 deletions and 3 insertions of single nucleotides (S1 Table). In four of 14 herds (Badlands National Park, National Elk Refuge, Tallgrass Prairie National Preserve, Theodore Roosevelt National Park) all 15 animals sampled had only Hap 1 sequences. Hap 5 was the second most frequent haplotype with two sub-haplotypes represented in 32 bison (15.3%) from 7 herds, with the highest frequency (10 of 15 animals) found in Grasslands National Park. All other haplotypes were present in 10 or fewer individuals across all herds. Six haplotypes (Hap 3, Hap 4, Hap 8, Hap 9, Hap 10, and Hap 11) were each found in only one herd, including the Book Cliffs (Hap 3 in one animal and Hap 4 with two sub-haplotypes in three animals), Grasslands National Park (Hap 8 in 3 animals), Rocky Mountain Arsenal National Wildlife Refuge (Hap 9 in 1 animal), Wind Cave National Park (Hap 10 with two mtDNA sub-haplotypes in two animals), and Wichita Mountains Wildlife Refuge (Hap 11 with two sub-haplotypes in 2 animals). Most herds were described by two or three mtDNA haplotypes, but we identified five mtDNA haplotypes (Hap 1, Hap 5, Hap 6, Hap7, and Hap 9) in the Rocky Mountain Arsenal bison herd.

Our median joining network of bison haplotypes reveals evolutionary relationships between mtDNA sequences (Fig 2). The network indicates that four haplotypes (Hap 2, Hap 8, Hap 9, and Hap 10, presenting in a star-like pattern) arose from the most frequent Hap 1, (centrally located within the star), through single mutation events. Hap 10 connects the rest of the network with most nodes separated by one mutation event, excluding Hap 4 and Hap 7 which underwent three mutation events. The right part reveals a number of intermediate, inferred haplotypes that were not identified in our study (Fig 2).

For the whole bison population sampled, the observed haplotype diversity was 0.48 ± 0.04 and the nucleotide diversity was 0.004 ± 0.002 (Table 2). Haplotype diversity among bison herds ranged from 0 for herds with only 1 haplotype identified to 0.81 in the Rocky Mountain Arsenal herd where 5 haplotypes were identified. Nucleotide diversity in the herds ranged

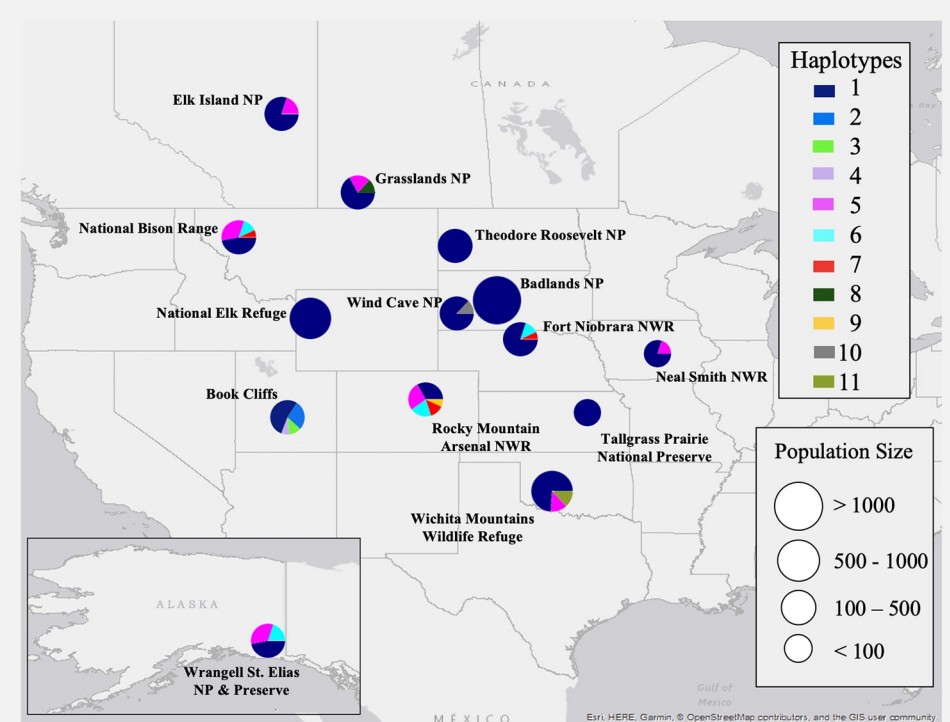

**Fig 1. Composition of mtDNA haplotypes (Hap 1 –Hap 11) detected in 14 U.S. Department of Interior and Parks Canada Agency bison herds.** Circle sizes are correlated with the size of each herd and the sectors in each chart correspond to percentage of mtDNA haplotypes. The map was created using ArcGIS® software by Esri (www.esri.com). Sources: Esri, HERE, Garmin, OpenStreetMap contributors, and the GIS User Community. Contains information from OpenStreetMap and OpenStreetMap Foundation, which is made available under the Open Database License.

from 0 in Badlands National Park, National Elk Refuge, and Theodore Roosevelt National Park to 0.006 in Rocky Mountain Arsenal, National Bison Range and Wrangell—St. Elias National Park and Preserve, each having several diverse haplotypes. The mean number of pairwise differences for all the bison herds is 3.38 ± 1.74 and varies from 0 for Badlands National Park, National Elk Refuge, and Theodore Roosevelt National Park to 5.68 in the Rocky Mountain Arsenal herd (Table 2).

A $F_{ST}$ genetic distance value of 0.21 (p = 0.000) confirmed some differentiation between the herds. Pairwise $F_{ST}$ values between populations ranged from zero to highest $F_{ST}$ values observed between Badlands National Park and Grasslands National Park bison ($F_{ST}$ = 0.61) and Grasslands National Park and National Elk Refuge bison ($F_{ST}$ = 0.61) (Fig 3). Likewise, AMOVA revealed more variation within bison herds (78.59%) than among herds (21.41%) (p <0.001) (S3 Table).

No statistical significance was detected for neutrality and population demographic increase, Tajima's D (Tajima's D = −0.34, p = 0.44) and Fu' test ($F_S$ = 2.16, p = 0.81), although the positive Fs is an indicator of haplotype deficiency expected after a recent bottleneck.

Pairwise differences between the major mtDNA haplotypes ranged from 0 to 7 substitutions (Fig 4). Raggedness index was 0.41 (p = 0.70), and the maximum nucleotide difference between 892 nucleotides of the control region was 0.78%.

MDS revealed small differences between populations in three clusters (Fig 5). National Elk Refuge, Badlands National Park, Tallgrass Prairie National Park, Theodore Roosevelt National

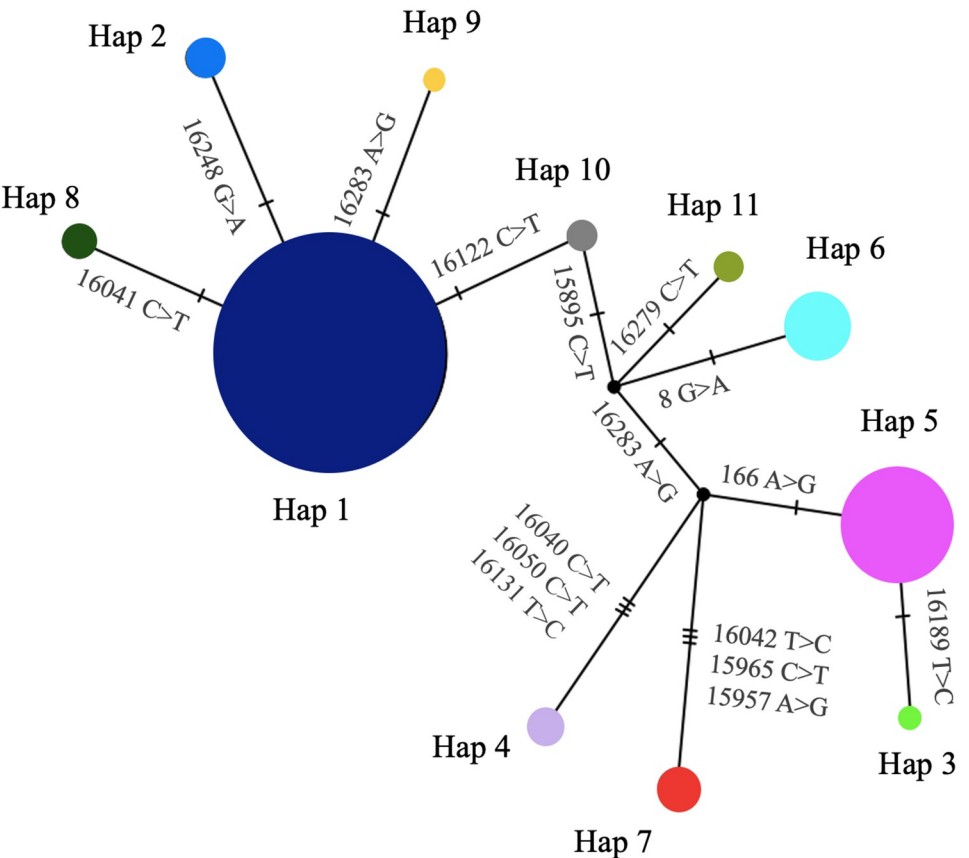

**Fig 2. Median joining network tree for bison haplotypes in 14 U.S. Department of Interior and Parks Canada Agency bison herds.** Circle size is proportional to haplotype frequency and hatch marks depict mutation events. Color of haplotype circles match those in Fig 1.

**Table 2. Summary population statistics of 14 U.S. Department of Interior and Parks Canada Agency bison herds, and all the herds assembled as a whole population.**

| Bison Herd | Haplotype Diversity | Nucleotide Diversity | Mean Pairwise Differences | No. Haplotypes |
|---|---|---|---|---|
| Badlands NP (BADL) | 0 | 0 | 0 | 1 |
| Book Cliffs (BOOK) | 0.68 ± 0.10 | 0.005 ± 0.003 | 4.63 ± 2.41 | 4 |
| Elk Island NP (ELK) | 0.34 ± 0.13 | 0.003 ± 0.002 | 2.74 ± 1.54 | 2 |
| Fort Niobrara NWR (FTN) | 0.36 ± 0.14 | 0.003 ± 0.002 | 2.99 ± 1.65 | 3 |
| Grasslands NP (GNP) | 0.53 ± 0.13 | 0.005 ± 0.003 | 4.49 ± 2.34 | 3 |
| National Bison Range (NBR) | 0.70 ± 0.08 | 0.006 ± 0.003 | 5.27 ± 2.70 | 4 |
| National Elk Refuge (NER) | 0 | 0 | 0 | 1 |
| Neal Smith NWR (NSM) | 0.34 ± 0.13 | 0.003 ± 0.002 | 2.74 ± 1.54 | 2 |
| Rocky Mountain Arsenal NWR (RMA) | 0.81 ± 0.06 | 0.006 ± 0.004 | 5.68 ± 2.88 | 5 |
| Tallgrass Prairie National Preserve (TAPR) | 0 | 0 | 0 | 1 |
| Wind Cave NP (WICA) | 0.25 ± 0.13 | 0.0006 ± 0.0006 | 0.50 ± 0.45 | 2 |
| Wichita Mountains Wildlife Refuge (WM) | 0.46 ± 0.15 | 0.004 ± 0.002 | 3.33 ± 1.80 | 3 |
| Wrangell St. Elias NP & Preserve (WRST) | 0.68 ± 0.17 | 0.006 ± 0.003 | 5.12 ± 2.63 | 3 |
| Theodore Roosevelt NP (TRNP) | 0 | 0 | 0 | 1 |
| All Populations | 0.48 ± 0.04 | 0.004 ± 0.002 | 3.38 ± 1.74 | 11 |

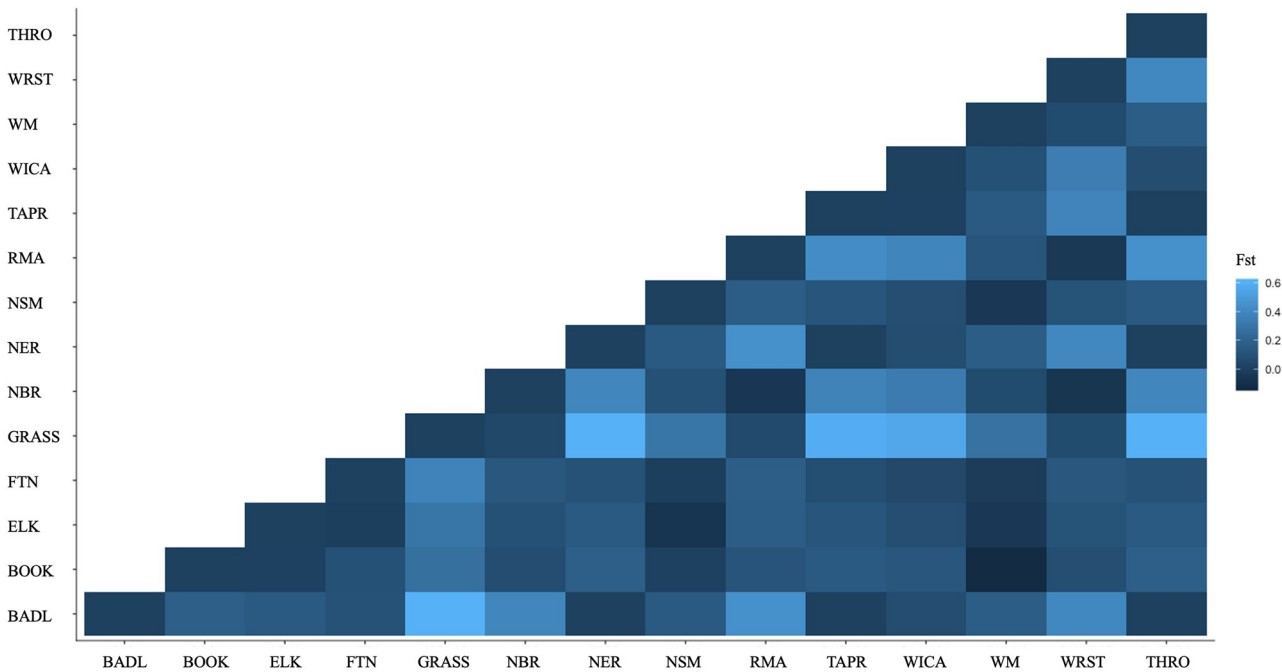

**Fig 3. Pairwise F$_{ST}$ matrix for 14 U.S. Department of Interior and Parks Canada Agency bison herds.** BADL, Badlands National Park; BOOK, Book Cliffs; ELK, Elk Island National Park; FTN, Fort Niobrara National Wildlife Refuge; GRASS, Grasslands National Park; NBR, National Bison Range; NER, National Elk Refuge/Grand Teton National Park; NSM, Neal Smith National Wildlife Refuge; RMA, Rocky Mountain Arsenal National Wildlife Refuge; TAPR, Tallgrass Prairie National Preserve; WICA, Wind Cave National Park; WM, Wichita Mountains Wildlife Refuge; WRST, Wrangell St. Elias National Park and Preserve; THRO, Theodore Roosevelt National Park.

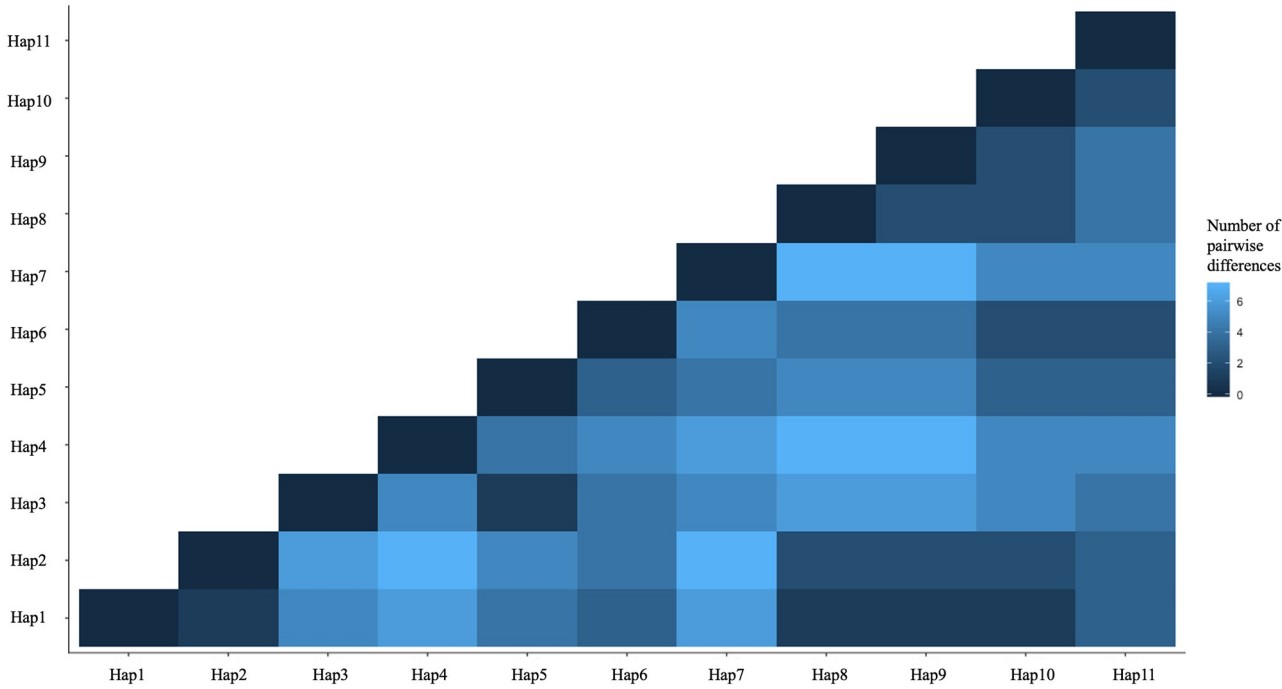

**Fig 4. Pairwise distance between the mtDNA control region haplotypes identified in 14 U.S. Department of Interior and Parks Canada Agency bison herds.**

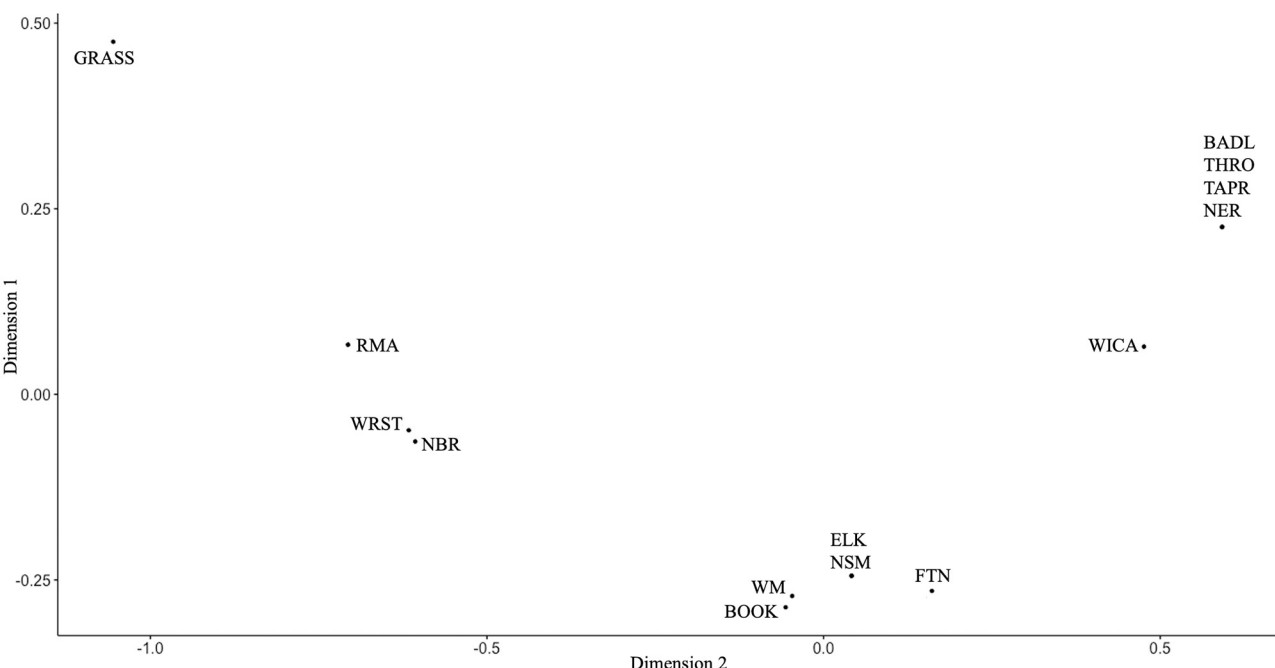

**Fig 5. Multi-dimensional scaling (MDS) plot of 14 U.S. Department of Interior and Parks Canada Agency bison herds based on $F_{ST}$ distances.**
BADL, Badlands National Park; BOOK, Book Cliffs; ELK, Elk Island National Park; FTN, Fort Niobrara National Wildlife Refuge; GRASS, Grasslands National Park; NBR, National Bison Range; NER, National Elk Refuge/Grand Teton National Park; NSM, Neal Smith National Wildlife Refuge; RMA, Rocky Mountain Arsenal National Wildlife Refuge; TAPR, Tallgrass Prairie National Preserve; WICA, Wind Cave National Park; WM, Wichita Mountains Wildlife Refuge; WRST, Wrangell St. Elias National Park and Preserve; THRO, Theodore Roosevelt National Park.

Park, and Wind Cave National Park samples form a small cluster likely due to the dominance of Hap 1 found in these herds. Book Cliffs, Fort Niobrara National Wildlife Refuge, Elk Island National Park, Neal Smith National Wildlife Refuge, and Wichita Mountain Wildlife Refuge samples formed another cluster with little difference in both dimensions. Within this cluster, the Elk Island National Park and Neal Smith NWR herds are not distinguished because both contain Hap 1 in 12 animals and Hap 5 in 3 animals and their Fs is zero. Rocky Mountain Arsenal National Wildlife Refuge, Wrangel—St. Elias National Park, and the National Bison Range samples formed another cluster, with all 3 sharing Haps 1, 5 and 6. Grasslands National Park is set apart from the rest of the bison herds in the MDS plot, likely because of the highest proportion of Hap 5 found in 10 animals and the unique Hap 8 in three animals (Fig 5).

## Discussion

Our study provides a modern update of mtDNA haplotype variation in plains bison conservation herds in North America, while offering an opportunity to compare the genetic and historic records detailing herd foundation and subsequent translocation events.

### Haplotype variation in plains bison conservation herds

While opportunities for comparison among mtDNA studies of bison have been limited due to designation of haplotypes mostly based on restriction sites and low quality of some mtDNA sequences [10], partial control region sequences [11] or the lack of the control regions in the mitochondrial genome sequences [19], we were able to compare the mtDNA control region haplotypes in our study with the complete control region sequences of bison mitochondrial

genomes published previously [12] (S4 Table). We found that four mtDNA control region haplotypes in our study are indistinguishable from several of Douglas et al., 2011 haplotypes [12] which were assigned based on different single nucleotide polymorphisms in the coding part of mitochondrial genome sequences. Our Hap 1 (including sub-haplotypes 1/0, 1/1, 1/7) is coincident with the control region sequences in bHap2, bHap 8, bHap10, bHap11, and bHap17; Hap 3 is consistent with bHap3 and bHap12; Hap 5 (5/2) is consistent with bHap4, bHap5, and bHap9; and Hap 7 is consistent with bHap13 and bHap16 (S4 Table). However, we did not find complete overlap between the two studies, as we did not detect the control region sequences of 3 previously identified haplotypes (bHap6, bHap7, wHap14) [12]. Additionally, seven of our haplotypes (Hap 2, Hap4, Hap 6, Hap 8, Hap 9, Hap 10, and Hap 11) did not match any sequences previously described, suggesting that both analyses have not captured all potentially existing global bison mtDNA diversity (S4 Table).

Consistent with the low mean difference of only 0.00103 across 10 haplotypes identified in 25 Yellowstone bison [19], we found low nucleotide diversity (0.004 ± 0.002) among 11 haplotypes identified in 209 bison from 14 herds. In contrast, Ward et al., 1999 [11] reported a nucleotide divergence of 9.91% ± 0.24% (or 0.0991 ± 0.0024) in a 677-bp section of the mtDNA control region between American bison and domestic cattle in 1999.

Recent common ancestry of modern bison deriving from small source populations and historic and recent admixture is reflected in Fst value, haplotype and nucleotide diversities and mean number of pairwise differences. Some herds, such as those at the Rocky Mountain Arsenal National Wildlife Refuge, Book Cliffs, and the National Bison Range, are relatively diverse, while others are almost homogenous (Badlands National Park, National Elk Refuge, and Theodore Roosevelt National Park). The genetic diversity is distributed unevenly among the conservation herds, suggesting clear areas for targeted mitochondrial gene flow.

The median joining network revealed two different patterns illustrating mtDNA evolution among extant bison. A star-like part of the network with the central Hap 1, consistent with recent population expansion from a small number of founders, indicates that mutational process generating genetic diversity may have occurred within herds since establishment. This mutational process is also suggested by the results of the small Tajima's D value, albeit statistically not significant. Branches of the network leading to Haps 4 and 7 reveal missing mutational steps in our data set, likely indicative of sampling bias, although loss through potential bottlenecks associated with initial demographic decline, perhaps in combination with subsequent isolation and drift, cannot be discounted. The ragged distribution of the observed number of pairwise differences across all mtDNA sequences indicated that many mtDNA sequences have few differences.

By comparing the composition of mtDNA haplotypes with previous work [12], we identified a larger diversity of mtDNA lineages in North American plains bison than previously reported, and a more extensive survey of mtDNA diversity in conservation herds will be necessary to achieve a complete understanding of the origin and importance of mitochondrial genetics for species conservation.

### Herd histories and phylogenetic relationships

Comparing the historic and genetic records of governmental conservation of bison herds provides additional insight into the results of our study. According to the chronological order of herd establishment, the Wichita Mountains Wildlife Refuge (WM) in Oklahoma, USA was one of the first federal herds founded in 1907 (Table 3). It was established with six bulls and nine cows selected by the New York Zoological Park Society in an attempt to represent at least 4 distinct original sources [20]. Four additional bulls were added from Fort Niobrara National

**Table 3. Establishment of bison herds.** Origin, founding members, additional introductions, and estimated size at sampling of 14 U.S. Department of Interior and Parks Canada Agency bison herds. Founding and translocation data from Coder, 1975 [24], Halbert and Derr, 2008 [21], Boyd, 2003 [23], Dratch and Gogan, 2010 [8], and Hartway et al., 2020 [9].

| Bison herd | Location | Year established | Foundation stock, sex composition | Source population | Additional introductions | | | Estimated herd size |
|---|---|---|---|---|---|---|---|---|
| | | | | | Number of animals | Source population | Year | |
| WM | USA, Oklahoma | 1907 | 6 bulls, 9 cows | New York Zoological Park | 4 bulls | Fort Niobrara NWR | 1940 | 576 |
| NBR | USA, Montana | 1909 | 13 bulls, 23 cows | Conrad Kalispel (Pablo-Allard) Herd, Montana | 2 bulls | 7-Up Ranch, Montana | 1939 | 302 |
| | | | 1 bull, 2 cows | Corbin (McKay-Alloway) Herd, New Hampshire | 4 bulls | Fort Niobrara NWR | 1952 | |
| | | | 1 cow | Charles Goodnight Herd, Texas | 2 bulls | Yellowstone NP | 1953 | |
| | | | | | 4 cows | Maxwell State Game Refuge (Jones), Kansas | 1984 | |
| ELK | Canada, Alberta | 1909 | 40–70 | Pablo-Allard Herd, Montana | | | | 470 |
| FTN | USA, Nebraska | 1913 | 1 bull, 5 cows | Gilbert herd, Nebraska | 4 bulls | Custer State Park | 1935 | 357 |
| | | | 2 bulls | Yellowstone NP | 4 bulls | Custer State Park | 1937 | |
| | | | | | 5 bulls | National Bison Range | 1952 | |
| | | | | | 1 bull | Wind Cave NP | 2010 | |
| | | | | | 39 | White Horse Hill National Game Preserve[1], North Dakota | 2010 | |
| | | | | | 4 bulls, 4 cows | Wichita Mountains Wildlife Refuge | 2011 | |
| WICA | USA, South Dakota | 1913 | 6 bulls, 8 cows | New York Zoological Park | 2 bulls, 4 cows | Yellowstone NP | 1916 | 350 |
| NER | USA, Wyoming | 1948 | 20 | Yellowstone NP | 6 bulls, 6 cows | Theodore Roosevelt NP | 1964 | 936 |
| WRST | USA, Alaska | 1950 | 17 | Delta herd[2], Alaska | | | | 181 |
| THRO (North and South) | USA, North Dakota | 1956 | 5 bulls, 24 cows (to South Unit) | Fort Niobrara NWR | 10 bulls, 10 cows (to North Unit) | Theodore Roosevelt NP (South Unit) | 1962 | 615 |
| BADL | USA, South Dakota | 1963 | 50 | Theodore Roosevelt NP | 20 | Colorado National Monument Herd | 1983 | 900 |
| | | | 3 | Fort Niobrara NWR | | | | |
| GRASS | Canada, Saskatchewan | 2005 | 30 bulls, 30 cows, 11 yearlings | Elk Island NP | | | | 346 |
| NSM | USA, Iowa | 2006 | 39 | National Bison Range | 2 bulls | Rocky Mountain Arsenal NWR | 2014 | 53 |
| RMA | USA, Colorado | 2007 | 16 | National Bison Range | 2 bulls | White Horse Hill National Game Preserve[1], North Dakota | 2008 | 71 |
| | | | | | 10 | National Bison Range | 2009 | |
| | | | | | 1 bull | Wind Cave NP | 2010 | |
| | | | | | 3 bulls | Wichita Mountains Wildlife Refuge | 2011 | |
| TAPR | USA, Kansas | 2009 | 13 | Wind Cave NP | 10 | Wind Cave NP | 2014 | 89 |

(*Continued*)

**Table 3.** (Continued)

| Bison herd | Location | Year established | Foundation stock, sex composition | Source population | Additional introductions | | | Estimated herd size |
|---|---|---|---|---|---|---|---|---|
| | | | | | Number of animals | Source population | Year | |
| **BOOK** | USA, Utah | 2009 | 14 | Ute Tribe Herd, Utah | 40 | Henry Mountains | 2010 | 540 |
| | | | 30 | Henry Mountains | | | | |

[1] The original White Horse Hill herd was established with bison from the Portland City Park herd, with subsequent introductions made from the National Bison Range, Fort Niobrara NWR, and Theodore Roosevelt NP. The current herd at White Horse Hill is entirely sourced from the National Bison Range.

[2] Animals of the National Bison Range origin.

WM, Wichita Mountains Wildlife Refuge; NBR, National Bison Range; ELK, Elk Island National Park; FTN, Fort Niobrara National Wildlife Refuge; WICA, Wind Cave National Park; NER, National Elk Refuge/Grand Teton National Park; WRST, Wrangell St. Elias National Park and Preserve; THRO, Theodore Roosevelt National Park; BADL, Badlands National Park; GRASS, Grasslands National Park; NSM, Neal Smith National Wildlife Refuge; RMA, Rocky Mountain Arsenal National Wildlife Refuge; TAPR, Tallgrass Prairie National Preserve; BOOK, Book Cliffs.

Wildlife Refuge in 1940 [21]. Two previous studies found two haplotypes in the Wichita herd [10, 11] while the present study found three mtDNA haplotypes determined by substitutions only and seven sub-haplotypes determined by substitutions and indels. Two of the three haplotypes found in the Wichita Mountains herd (Hap 1 and Hap 5) are the most common in other bison populations within this study, while Hap 11 was not found in any other herd. Given the limited augmentation history for this herd, the origin of this unique haplotype is most likely attributed to the diversity of founders as described in the selection of animals by the New York Zoological Society for shipment to Wichita Mountains in 1907.

The National Bison Range (NBR) herd in Montana, USA was founded in 1909. Thirteen bulls and twenty-three cows came from the Conrad herd of Pablo-Allard foundation, with one bull and two cows from the Corbin herd, and one cow from Charles Goodnight [20]. In 1939, two bulls were added from the 7-Up Ranch in Montana. Ten more individuals were introduced, including four bulls from Fort Niobrara National Wildlife Refuge in 1952, two bulls from Yellowstone National Park in 1953, and four cows from the Maxwell State Game Refuge in 1984 [22] (Table 3). Previous studies found four haplotypes [10] and three haplotypes [11], respectively. We observed four haplotypes (Hap 1, Hap 5, Hap 6, and Hap 7) and the second largest pairwise differences (0.70 ± 0.08) that make the National Bison Range one of the most diverse conservation populations. Establishment with females from 3 original foundation herds (Pablo-Allard, Corbin and Goodnight), combined with subsequent augmentation of cows descended from at least one additional early, distinct origin source (Jones herd via Maxwell State Game Refuge, Kansas), likely explains the high levels of observed mtDNA diversity.

In 1909, Elk Island National Park (ELK) herd was established in Alberta, Canada with 40–70 bison from the Pablo-Allard foundation herd [9, 23]. Polziehn et al., 1996 [10] observed five haplotypes from 40 plains bison; however, it is difficult to interpret and evaluate these results because Polziehn et al., 1996 [10] used two different restriction digestion techniques. In agreement with our study, Ward et. al., 1999 [11] found two haplotypes in a sample of 19 plains bison. Hap 1 is ubiquitous among herds in this study, and along with Hap 5 occurred in several of the earliest established federal conservation herds. Hap 5 occurs in both Elk Island National Park (direct and exclusive Pablo-Allard origin) and Wichita Mountains National Wildlife Refuge (possibly through the Pablo-Allard contribution to the New York Zoological Park herd). With no augmentation of cows after initial herd establishment in either of these 2 herds, along with its presence in the National Bison Range herd (also established early and primarily from

Pablo-Allard foundation), Hap 5 is likely to be of original Pablo-Allard lineage, reflecting diversity that existed in North American plains bison prior to the bottleneck (Table 3).

The Fort Niobrara National Wildlife Refuge herd (FTN) in Nebraska, USA was founded in 1913 from one bull and five cows sourced from the privately-owned Gilbert herd in Nebraska, and two bulls from Yellowstone National Park [22, 24]. Four bulls from Custer State Park in South Dakota were added in 1935, and they would also contribute four more bulls in 1937 [22]. In 1952, five bulls were added from the National Bison Range [22], and a bull originating from Wind Cave National Park was added in 2010 [9]. In 2010, thirty-nine mixed sex and age bison were also introduced from White Horse Hill National Game Preserve in North Dakota and in 2011 four bulls and four cows were transferred to Fort Niobrara from the Wichita Mountains National Wildlife Refuge [9] (Table 3). Fort Niobrara has the most introductions and has also contributed to several other populations in this study (Badlands National Park, Theodore Roosevelt National Park, National Bison Range, National Elk Refuge and Wichita Mountains Wildlife Refuge), albeit genetic contributions made through translocation of bulls would not be identified in this study. While previous mtDNA studies of the Fort Niobrara bison herd found one haplotype [10, 11], our work identified three haplotypes (Hap 1, Hap 6, Hap 7). (Table 3; S2 Table). Hap 6 and Hap 7 are shared only by the National Bison Range, Fort Niobrara, and Rocky Mountain Arsenal National Wildlife Refuge herds. It is unknown whether these 2 haplotypes were present in the early founders of both National Bison Range and Fort Niobrara herds, or if they were introduced into the Fort Niobrara herd when combined with the original White Horse Hill National Game Preserve herd. The original White Horse Hill herd, reported in previous studies using its former name Sully's Hill, was reported to have originally been sourced from Ravalli, Montana for the Portland City Park in Oregon [8], and may have included Pablo-Allard origin animals based on geographic proximity (Table 3). Although bison were also exchanged between the National Bison Range and Fort Niobrara back in 1952, only bulls were reported to have been included in each of those two transfers [22]. Introductions from Yellowstone into both National Bison Range and Fort Niobrara herds were also bulls, suggesting that Hap 6 and Hap 7 are also original foundation haplotypes that were captured during the establishment of herds early in the conservation era.

The Wind Cave National Park (WICA) bison herd in South Dakota, USA, established in 1913, originated from six bulls and eight cows from the New York Zoological Park and was supplemented with 2 bulls and 4 cows from Yellowstone National Park in 1916 [21]. In 1964 and 1979, the herd was reduced drastically in an effort to eradicate brucellosis [8]. Ward et al., 1999 [11] found one haplotype compared to the two haplotypes found in this study (Hap 1 and Hap 10) (Fig 2). We found Hap 10 unique to Wind Cave bison, possibly representing variation that existed in the first conservation herds (e.g., the New York Zoological Park or the remnant herd in Yellowstone National Park), with mtDNA lineages from pre-extermination North American plains bison populations.

The National Elk Refuge (NER) herd in Wyoming, USA was established in 1948 with twenty bison from Yellowstone National Park, but the herd was reduced to 9 animals in 1963 in an effort to eradicate brucellosis. Six bulls and six cows from Theodore Roosevelt National Park were added in 1964 [22] (Table 3). This migratory herd has a diverse management history but is now co-managed across multiple jurisdictions by the NPS (Grand Teton National Park) and FWS, in partnership with Wyoming Game and Fish Department. There is no previous information for mtDNA genetic diversity within the National Elk Refuge bison, but our findings indicate low diversity, with only Haplotype 1 observed. Although sample size in our study could potentially limit detection of additional haplotypes, such low diversity is consistent with the historic management of this herd, including a significant reduction in numbers followed by decades of little or no population growth until the mid-1990s.

The Wrangell—St. Elias National Park and Preserve (WRST) Copper River bison herd in Alaska, USA was established in 1950 with seventeen mixed sex animals transferred from the Delta herd in Alaska [23] which was founded from 23 bison from the National Bison Range in 1928 [9] (Table 3). Three haplotypes were identified in this study (Hap 1, Hap 5, and Hap 6), all shared with the source National Bison Range herd and likely to be of original foundation lineages. However, the National Bison Range herd carries Hap 5/2 while the Wrangell—St. Elias herd has Hap 5/1 distinguishing 221.1C in Hap 5/1. This variation can be explained by incomplete sampling of mtDNA lineages in National Bison Range or de novo mutation in one of these herds as 221.1C is a frequent insertion found across several haplotypes (Hap1, Hap 5, Hap 8).

In 1956, Theodore Roosevelt National Park (THRO) in North Dakota, USA established their South Unit bison herd with five bulls and twenty-four cows from the Fort Niobrara National Wildlife Refuge. In 1962, ten bulls and ten cows were taken from the south unit to establish the park's north unit herd (Table 3). We detected one haplotype (Hap 1 with sub-haplotypes Hap 1/0, Hap 1/4, and Hap 1/7) in a mix of bison from the North and South Unit populations, in contrast to identifying only one sub-haplotype (Hap 1/1) in the Fort Niobrara National Wildlife Refuge founder herd (S1 Table). This dissimilarity can be explained by incomplete sampling in the Fort Niobrara herd, or alternatively by the potential for mutation processes occurring in Theodore Roosevelt National Park bison.

The Badlands National Park (BADL) herd in South Dakota, USA was founded in 1963 with 50 bison from Theodore Roosevelt National Park and 3 from Fort Niobrara National Wildlife Refuge. In 1983, twenty more bison were added from Colorado National Monument when the Monument herd was disbanded in the1980's due to a lack of grazing resources on the preserve [22] (Table 3). Only Hap 1 (Hap 1/0) was found in Badlands National Park, consistent with the haplotype compositions in the source populations of Theodore Roosevelt National Park and Fort Niobrara National Wildlife Refuge.

In 2005, the Grasslands National Park (GRASS) in Saskatchewan, Canada brought in 30 bulls, 30 cows, and 11 yearlings from Elk Island National Park to establish a new herd (Table 3). Grasslands National Park bison share the same two haplotypes Hap 1 (Hap 1/1) and Hap 5 (Hap 5/1 and Hap 5/2) with the source herd at Elk Island. Unique Hap 8 –not detected in any other Pablo-Allard foundation herd–likely signals either emerging diversity in the Grasslands National Park population through mutational process or an undetected haplotype derived from the Elk Island National Park herd. While the MDS analysis identifies Grasslands National Park as the most separated from all other herds due to highest proportion of Hap 5 found in 10 animals and unique Hap 8 in three animals (Fig 5), given its direct foundation from Elk Island in modern times, this apparent distance almost certainly reflects either founder effect or sampling bias.

Neal Smith National Wildlife Refuge (NSM) in Iowa, USA established a new population in 2006 with thirty-nine bison from the National Bison Range [8]. In 2014, the National Bison Range origin herd at the Rocky Mountain Arsenal National Wildlife Refuge contributed two bulls (Table 3). We found two National Bison Range origin haplotypes (Hap 1 and Hap 5) in Neal Smith National Wildlife Refuge bison, while two additional haplotypes present at the National Bison Range were either lost at Neal Smith due to founder effect or were not detected in this study due to sampling artifact. However, sub-haplotype 5/1 determined by the presence of 221.1C was not found in National Bison Range or Rocky Mountain Arsenal National Wildlife Refuge bison, where sub-haplotype 5/2 was instead identified. This difference can be explained by incomplete sampling of mtDNA lineages in any of the National Bison Range origin source herds, or de novo sub-haplotype mutation in bison with Hap 5 at Neal Smith National Wildlife Refuge.

The Rocky Mountain Arsenal National Wildlife Refuge (RMA) in Colorado, USA transferred 16 bison from the National Bison Range to establish this FWS managed herd in 2007. A year later, two bulls were added from the National Bison Range origin herd established at White Horse Hill [8], with 10 additional animals added from the National Bison Range in 2009, followed by a Wind Cave National Park origin bull in 2010 [9]. Wichita Mountains contributed three bulls in 2011 [9] (Table 3). The Rocky Mountain Arsenal herd is the most diverse in this study, with five haplotypes (Hap 1, Hap 5, Hap 6, Hap 7, Hap 9) observed, and haplotype diversity and mean pairwise values of 0.81 ± 0.06 and 5.68 ± 2.88, respectively. Hap 1, Hap 5, Hap 6, and Hap 7, original foundation lineages reflecting pre-bottleneck diversity, are consistent with National Bison Range origin. Hap 9 is unique to the Rocky Mountain Arsenal National Wildlife Refuge population, signaling either emerging diversity due to mutational process within the refuge or additional undetected haplotypes that likely exist in the National Bison Range herd.

The Tallgrass Prairie National Preserve (TAPR) in Kansas, USA established a bison population in 2009 with 13 animals from Wind Cave National Park. An additional 10 Wind Cave animals were added in 2014, and the Tallgrass Prairie herd serves as a satellite herd for Wind Cave National Park [9] (Table 3). Hap 1 is the only haplotype identified in Tallgrass Prairie bison with four sub-haplotypes (Hap 1/0, Hap 1/1, Hap 1/3, and Hap 1/5). The sub-haplotype composition is an agreement with the Wind Cave National Park herd having mostly Hap 1/0 and Hap 1/1. The presence of Hap 1/3 and Hap 1/5 can be explained by incomplete sampling in Wind Cave National Park or de novo mutations in the Tallgrass Prairie National Preserve herd.

Book Cliffs (BOOK) in Utah, USA established a bison herd in 2009 with 14 bulls and cows from the Ute Tribe herd in Utah and 30 bison from Henry Mountains [24, 25]. In 2010, an additional 40 animals from the Henry Mountains herd were introduced [9] (Table 3). The Ute Tribe herd was sourced from multiple, unknown herds, while Henry Mountains bison originated from a founder group exported from Yellowstone National Park in the 1940s [9, 26]. We found four haplotypes (Hap 1, Hap 2, Hap 3, Hap 4) represented in this herd, three of which are unique (Hap 2, Hap 3, Hap 4). This surprising discovery in the most recent conservation herd most likely confirms the existence of additional, not yet characterized diversity of the mtDNA control region in Yellowstone National Park, or other unknown sources potentially contributing to the Ute Tribe herd over time. Although mtDNA cattle introgression the Book Cliffs herd has been confirmed in previous studies, we selected samples from animals determined to be non-introgressed.

## Limitations and implication for species conservation

Our study has several limitations, including certain potential for sampling bias. For example, while the Rocky Mountain Arsenal herd recently established with National Bison Range founders captures all NBR mtDNA haplotypes, the lower diversity at Neal Smith National Wildlife Refuge indicates either founder effect with an apparent failure to capture Hap 6 and 7 during establishment, or failure to detect them in this study due to relatively small sample size. Similarly, either founder effect or sampling bias is likely when comparing Wind Cave National Park and Tallgrass Prairie herds as well as considering the unique haplotype (Hap 8) found in Grasslands National Park that was not detected in its source herd, Elk Island. MDS analysis illustrated these limitations in interpretation, which may be resolved by future work addressing samples size, as well as targeting samples collected from animals known to be born prior to specific translocation events.

However, despite these limitations, we found strong correlation between our genetic analysis and historic records of the bison conservation herds in the U.S. and Canada. Additionally,

we were able to identify the potential for evolutionary forces in the form of mutational process, which may continue to generate new genetic variation at some unknown rate. Finally, we provide an improved characterization of the mtDNA diversity present in the earliest conservation herds that served to preserve and support both genetic and demographic recovery of this species over the past century.

Future comparisons between haplotype diversity including whole mitochondrial genome sequences, the maternal lineage composition of bison populations, and herd management practices may help inform the next generation of bison managers in their efforts to ensure long-term species viability. Despite limitations caused by possible sampling bias in our study, mtDNA diversity may have been more intensely impacted by bottlenecks, founder effect, and genetic drift than reported for nuclear diversity [9], with fewer surviving matrilines in populations and a trend towards homogeneity in some extant herds. Simple evaluation of geographic distribution of existing mtDNA variation (Fig 1) reveals disproportionate diversity across herds, with some haplotypes occurring in only one location and some herds exhibiting monoclonal mtDNA profiles. To ensure conservation of global mitochondrial diversity and improve population genetics, gene flow strategies should seek to 1) increase mitochondrial diversity in herds with few haplotypes, 2) transfer rare haplotypes (e.g., Haps 4, 8–11) to other herds so their occurrence is not dependent on the survival of a single population, and 3) include sources of multiple mitochondrial lineages when planning for development of new herds. Moving forward, bison managers should include considerations for conservation, redistribution, and banking of mitochondrial lineages alongside strategies identified to manage nuclear genetic diversity [9], to ensure that existing variation persists and is preserved as a long-term objective for bison conservation.

## Supporting information

**S1 Table. MtDNA control region haplotypes determined by nucleotide substitutions and indels found in 14 U.S. Department of Interior (DOI) and Parks Canada Agency (PCA) bison herds.** Polymorphisms were determined by alignment with the reference mitochondrial genome GU946990.
(DOCX)

**S2 Table. Distribution of the mtDNA control region haplotypes determined by nucleotide substitutions and indels among 14 U.S. Department of Interior (DOI) and Parks Canada Agency (PCA) bison herds.**
(DOCX)

**S3 Table. Analysis of Molecular Variance (AMOVA) for 14 U.S. Department of Interior (DOI) and Parks Canada Agency (PCA) bison herds using pairwise differences.**
(DOCX)

**S4 Table. Comparison of mtDNA control region haplotypes found in 14 U.S. Department of Interior (DOI) and Parks Canada Agency (PCA) bison herds in our study with the mtDNA sequences published in Douglas et al., 2011 [12].**
(DOCX)

## Acknowledgments

We thank two Reviewers for providing constructive and valuable comments that helped us to improve the manuscript.

## Author Contributions

**Conceptualization:** Blake McCann, Igor V. Ovchinnikov.

**Data curation:** Maria Cecilia Penedo, Igor V. Ovchinnikov.

**Formal analysis:** Gaimi Davies, Maria Cecilia Penedo.

**Funding acquisition:** Blake McCann, Igor V. Ovchinnikov.

**Investigation:** Gaimi Davies, Lee Jones, Maria Cecilia Penedo.

**Methodology:** Maria Cecilia Penedo, Igor V. Ovchinnikov.

**Resources:** Blake McCann, Lee Jones, Stefano Liccioli.

**Visualization:** Gaimi Davies.

**Writing – original draft:** Gaimi Davies, Blake McCann, Maria Cecilia Penedo.

**Writing – review & editing:** Blake McCann, Lee Jones, Stefano Liccioli, Maria Cecilia Penedo, Igor V. Ovchinnikov.

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
