## [Decision Letter · Decision Letter 0]

16 Jun 2021

PONE-D-21-13309

Genetic variation of the mitochondrial DNA control region across plains bison herds in USA and Canada

PLOS ONE

Dear Dr. Ovchinnikov,

Thank you for submitting your manuscript to PLOS ONE. After careful consideration, we feel that it has merit but does not fully meet PLOS ONE’s publication criteria as it currently stands. Therefore, we invite you to submit a revised version of the manuscript that addresses the points raised during the review process.

Both reviewers considered your study has high quality and relevance for conservation and suggest some particular revisions to certain points, including your suggestions for the conservations of these herds.

We look forward to receiving your revised manuscript.

Kind regards,

Susana Caballero, PhD

Academic Editor

PLOS ONE

2. We note that Figure 1 in your submission contain [map/satellite] images which may be copyrighted. All PLOS content is published under the Creative Commons Attribution License (CC BY 4.0), which means that the manuscript, images, and Supporting Information files will be freely available online, and any third party is permitted to access, download, copy, distribute, and use these materials in any way, even commercially, with proper attribution. For these reasons, we cannot publish previously copyrighted maps or satellite images created using proprietary data, such as Google software (Google Maps, Street View, and Earth). For more information, see our copyright guidelines: http://journals.plos.org/plosone/s/licenses-and-copyright.

In the figure caption of the copyrighted figure, please include the following text: “Reprinted from [ref] under a CC BY license, with permission from [name of publisher], original copyright [original copyright year].

Additional Editor Comments (if provided):

Both editors agree that this is a good quality study presenting important and interesting information. They have some minor questions that the authors should clarify in the text for increased clarity for the readers. Also, in the introductions, the authors mention they will provide recommendations for conservation of the bison herds, but they do not include this in the discussion. I would suggest the authors to include these conservation applications of your study.

Reviewers' comments:

Reviewer's Responses to Questions

**Comments to the Author**

1. Is the manuscript technically sound, and do the data support the conclusions?

Reviewer #1: Partly

Reviewer #2: Yes

2. Has the statistical analysis been performed appropriately and rigorously? 

Reviewer #1: Yes

Reviewer #2: Yes

3. Have the authors made all data underlying the findings in their manuscript fully available?

Reviewer #1: Yes

Reviewer #2: Yes

4. Is the manuscript presented in an intelligible fashion and written in standard English?

Reviewer #1: Yes

Reviewer #2: Yes

5. Review Comments to the Author

Reviewer #1: Genetic variation of the mitochondrial DNA control region across plains bison herds in USA and Canada

The authors describe the haplotypic diversity of the mitochondrial DNA control region of 14 plains bison herds in the US and Canada. They describe the diversity within and between herds, and compare these haplotypes to the historical record. The authors conclude that more thorough sampling will be required because they both identified novel haplotypes and were unable to find previously described haplotypes.

Although the discussion of the herd histories is interesting, the introduction states that the authors will "provide recommendations for herd conservation management based on our analysis" but I don't see this present in those discussions.

lines 134-135 - it is unclear to me if none of your bison sampled contained cattle mtDNA or if you found but excluded samples with cattle mtDNA. These makes it sound like you didn't find any but lines 316-317 indicate that animals with cattle mtDNA were excluded. Please clarify.

line 200 - change "10 or less" to "10 or fewer"

lines 316-317- it would be interesting to know how many animals with cattle mtDNA were excluded. Did you find fewer animals with cattle mtDNA than previous studies? I know that identifying such introgressions was important in the past.

Reviewer #2: Davies and colleagues carried out extensive sampling of 14 bison conservation herds across the US and Canada in order to characterize current mitochondrial control region diversity within and among these herds. They also considered their findings in the context of past studies on bison mitochondrial genetic diversity and historical records of the founding and subsequent management of bison conservation herds.

To assess mitochondrial control region diversity, they extracted DNA from biopsies and hairs taken from bison and used Sanger sequencing to generate control region sequences from a total of 209 bison across the 14 herds. They discovered 11 haplotypes distributed unequally across herds, with the most common haplotype found among all herds, while several haplotypes were found only in single herds. They suggest that genetic differentiation is low between conservation herds, likely due to limited genetic diversity remaining from their recent population bottleneck and to the common origins of several herds. and find some evidence of genetic effects of the recent bottleneck and possibly of the recent expansion following the population collapse.

The study has employed what appeared to be a carefully chosen sampling scheme and their analysis tools and decisions seem appropriate for the data they generated. I think that this study provides new information about the existing genetic diversity of bison conservation herds and will potentially be considered by those managing herds in order to take into account the genetic diversity present across herds.

I have only several minor comments, listed below on a point-by-point basis, which the authors may want to address.

Minor comments:

Page 4, line 36: Here and elsewhere the authors characterize genetic summary statistics, such as Fst and nucleotide diversity, of the bison CR examined as low, but it’s unclear to me what comparisons or standards are being used to define what low diversity is in this case. Is Fst/nucleotide diversity low in comparison to past bison genetic diversity, or in comparison to other similar species? It may be helpful to contextualize low here or simply rely on the numeric descriptors instead.

Page 7, line 137: Is there a citation or other reference available for the specific extraction protocols used?

Page 8, line 166: Perhaps it would also be useful to mention raggedness index here, if Arlequin was also used for this analysis.

Page 8, line 172: It’s slightly unclear to me what is being used as input for the MDS analysis. The authors state that “Genetic distances used for MDS were based on pairwise FST values between populations,” so are the genetic distances simply the matrix of Fst values between populations, or are they being transformed in some way prior to MDS?

Page 12, line 224: Wouldn’t the Fst be the same for any comparison between Grasslands NP and any of the herds where all sampled bison had Haplotype 1, rather than just the two mentioned?

Page 14, line 269: The authors state that MDS reveals only small differences between herds, though I’m not sure this is correct, since MDS as presented isn’t really quantifying the extent of difference but rather showing major axes of variation. Perhaps instead it might be clearer to state something along the lines of “MDS doesn’t reveal strong clustering structure,” if this is what is meant by the authors? However, there does seem to be at least three distinct clusters, as described by the authors.

Page 16, line 314: Perhaps be “low mean”, instead of “mean low”?

Page 17, line 337: The authors suggest that the raggedness index suggests a constant population size throughout the 20th century for the bison metapopulation. However, do we know from the historical record that this isn’t true, with the population increasing throughout the 20th century, particularly at the beginning?

Page 25, line 482: The authors mention sampling bias here and later in the discussion. I wonder if it might be possible to show a rarefaction curve of discovered haplotypes with increasing sample numbers or something similar to address how much of the total haplotypic diversity has been sampled and how many new haplotypes would likely be discovered with increased sampling.

As a final note, it would be great if the authors could indicate where and how the data will be made available (presumably on GenBank?).

6. PLOS authors have the option to publish the peer review history of their article (what does this mean?). If published, this will include your full peer review and any attached files.

Reviewer #1: No

Reviewer #2: **Yes: **Jonas Oppenheimer

---

## [Author Response · Author response to Decision Letter 0]

15 Oct 2021

REBUTTAL LETTER

PLoS One resubmission

Manuscript PONE-D-21-13309R1

Genetic variation of the mitochondrial DNA control region across plains bison herds in USA and Canada

Gaimi Davies, Blake McCann, Lee Jones, Stefano Liccioli, Maria Cecilia Penedo, Igor V. Ovchinnikov

Response to the Academic Editor and Reviewers

2. Figure 1 in the submission contains [map/satellite] images which may be copyrighted.

The Figure 1 image was created by an author who used a base map in ArcGIS Online map hosted by Esri to create a static map (www.esri.com). The terms of use for static maps are included below. 

Terms of use for static maps (https://doc.arcgis.com/en/arcgis-online/reference/static-maps.htm#ESRI_SECTION1_21347CA4FAB14A7E95CE6B738DCA2843):

“You can use static maps that include ArcGIS Online maps hosted by Esri in printed or digital reports and PowerPoint presentations for colleagues, subsidiaries, and customers. A static map includes screen captures, a printed or plotted map, a map used in a PDF file, a map used in a PowerPoint presentation, or any other static rendering of a map.

The following uses are permitted:

• Personal use, internal business use, or to include in a presentation or a report for a client

• In brochures and marketing collateral, or on a company website to promote your own products and services and display your store locations

• In academic publications (for example, research journals, textbooks, and so on).”

2a. You may seek permission from the original copyright holder of Figure 1 to publish the content specifically under the CC BY 4.0 license.

The Esri terms of use for static maps show that we do not need to seek permission from the original copyright holder of the map in Figure 1. This basemap layer, created by ESRI and used in ArcGIS, is able to be built upon by users and saved as their own map. This map in its altered form, with our data overlayed on top of the "Light Gray Canvas Basemap", is able to be used in research publications as long as the base map is properly credited within the publication.

We added the sentence “Sources: Esri, HERE, Garmin, OpenStreetMap contributors, and the GIS User Community. Contains information from OpenStreetMap and OpenStreetMap Foundation, which is made available under the Open Database License.” to the Fig 1 legend, Lines 214 – 217 in the revised version of the manuscript. 

Please review your list reference list to ensure that it is complete and correct. Any changes to the reference list should be mentioned in the rebuttal letter that accompanies your revised manuscript.

The reference list is complete and correct. 

We added a reference [13] for the specific extraction protocol answering the minor comment of Reviewer #2 (Page 7, line 137), changing a sentence

“DNA was extracted from 5-6 hair roots or from 3 mm punches of skin tissue using standard Proteinase-K digestion protocols.”

to 

“DNA was extracted from 5-6 hair roots or from 3 mm diameter punches of skin tissue with a standard Proteinase-K digestion protocols used by the Veterinary Genetics Laboratory, U.C. Davies [13].” in Lines 136 – 138 in the revised version of the manuscript.

The full reference [13] was added to References in Lines 605 – 606:

Locke MM, Penedo MC, Bricker SJ, Millon LV, Murray JD. Linkage of the grey coat colour locus to microsatellites on horse chromosome 25. Anim Genet. 2002;33: 329-337.

The addition of the reference [13] to the reference list shifted the numbers of the original references [13] – [25] to [14] – [26].

Due to the addition of the reference [13], we re-numbered the next references in the manuscript text:

Line 159. Changed [13] to [14].

Line 163. Changed [14] to [15].

Line 167. Changed [15] to [16].

Line 174. Changed [16] to [17].

Line 176. Changed [17] to [18].

Line 303. Changed [18] to [19].

Line 318. Changed [18] to [19].

Line 352. Changed [19] to [20].

Line 353. Changed [20] to [21].

Line 363 (Table 3 legend). Changed [23] to [24], [20] to [21], [22] to [23].

Line 377. Changed [19] to [20].

Line 381. Changed [21] to [22].

Line 389. Changed [22] to [23].

Line 403. Changed [21] to [22] and [23] to [24].

Line 404. Changed [21] to [22].

Line 405. Changed [21] to [22].

Line 424. Changed [21] to [22].

Line 430. Changed [20] to [21].

Line 440. Changed [21] to [22].

Line 450. Changed [22] to [23].

Line 471. Changed [21] to [22].

Line 518. Changed [23] to [24] and [24] to [25].

Line 521. Changed [25] to [26]. 

Line 607 (in References). Changed [13] to [14].

Line 609 (in References): Changed [14] to [15].

Line 611 (in References): Changed [15] to [16].

Line 614 (in References): Changed [16] to [17].

Line 616 (in References): Changed [17] to [18].

Line 618 (in References): Changed [18] to [19].

Line 620 (in References): Changed [19] to [20].

Line 622 (in References): Changed [20] to [21].

Line 624 (in References): Changed [21] to [22].

Line 628 (in References): Changed [22] to [23].

Line 631 (in References): Changed [23] to [24].

Line 636 (in References): Changed [24] to [25].

Line 638 (in References): Changed [25] to [26].

Additional Editor Comments: 

In the introductions, the authors mention they will provide recommendations for conservation of the bison herds, but they do not include this in the discussion. I would suggest the authors to include these conservation applications of your study.

Reviewer #1 in Section 5. Review Comments to the Author: The introduction states that the authors will “provide recommendations for herd conservation management based on our analysis” but I don’t see this present in those discussions.

We added recommendations in the end of Discussion in Lines 555 – 566:

Simple evaluation of geographic distribution of existing mtDNA variation (Fig 1) reveals disproportionate diversity across herds, with some haplotypes occurring in only one location and some herds exhibiting monoclonal mtDNA profiles. To ensure conservation of global mitochondrial diversity and improve population genetics, gene flow strategies should seek to 1) increase mitochondrial diversity in herds with few haplotypes, 2) transfer rare haplotypes (e.g., Haps 4, 8 – 11) to other herds so their occurrence is not dependent on the survival of a single population, and 3) include sources of multiple mitochondrial lineages when planning for development of new herds. Moving forward, bison managers should include considerations for conservation, redistribution, and banking of mitochondrial lineages alongside strategies identified to manage nuclear genetic diversity [9], to ensure that existing variation persists and is preserved as a long-term objective for bison conservation.

In connection to the above change, Lines 552 – 556 in the original manuscript were modified from 

“However, in spite of limitations caused by possible sampling bias in our study, mtDNA diversity due to the mtDNA maternal inheritance promoting mtDNA homogeneity, may have been more intensely impacted by bottlenecks, founder effect, and genetic drift than reported for nuclear diversity [9], resulting in monoclonal mtDNA profiles in some herds.”

to 

“Despite limitations caused by possible sampling bias in our study, mtDNA diversity may have been more intensely impacted by bottlenecks, founder effect, and genetic drift than reported for nuclear diversity [9], with fewer surviving matrilines in populations and a trend towards homogeneity in some extant herds.” in Lines 552 – 555 in the revised version of the manuscript.

We also deleted a sentence in Lines 556 – 558 in the original manuscript:

“Moving forward, bison managers should include considerations for conservation, redistribution, and banking of mitochondrial lineages along with nuclear genetics, to ensure that existing variation persists and is preserved as a long-term objective for bison conservation.”

Reviewer #1 in Section 5. Review Comments to the Author:

lines 134-135 - it is unclear to me if none of your bison sampled contained cattle mtDNA or if you found but excluded samples with cattle mtDNA. These makes it sound like you didn't find any but lines 316-317 indicate that animals with cattle mtDNA were excluded. Please clarify.

As stated in Lines 134-135, “all the selected samples had bison-origin mtDNA, with no evidence of cattle introgression.”

We modified Lines 314-317 in the original manuscript “Consistent with the mean low difference of only 0.00103 across 10 haplotypes identified in 25 Yellowstone bison [18], we found low nucleotide diversity (0.004 ± 0.002) among 11 haplotypes identified in 209 bison from 14 herds from which we excluded animals with haplotypes previously identified as of cattle origin.” 

to 

“Consistent with the low mean difference of only 0.00103 across 10 haplotypes identified in 25 Yellowstone bison [19], we found low nucleotide diversity (0.004 ± 0.002) among 11 haplotypes identified in 209 bison from 14 herds.” in Lines 317 – 319 in the revised version of the manuscript.

line 200 - change "10 or less" to "10 or fewer"

Line was changed from “All other haplotypes were present in 10 or less individuals across all herds.” 

to 

“All other haplotypes were present in 10 or fewer individuals across all herds.” in Lines 201 – 202 in the current version of the manuscript.

lines 316-317- it would be interesting to know how many animals with cattle mtDNA were excluded. Did you find fewer animals with cattle mtDNA than previous studies? I know that identifying such introgressions was important in the past.

None of our samples contained cattle mtDNA. This line was corrected in Lines 317 – 319 in the revised version of the manuscript as mentioned above. 

Reviewer #2 in Section 5. Review Comments to the Author:

Page 4, line 36: Here and elsewhere the authors characterize genetic summary statistics, such as Fst and nucleotide diversity, of the bison CR examined as low, but it’s unclear to me what comparisons or standards are being used to define what low diversity is in this case. Is Fst/nucleotide diversity low in comparison to past bison genetic diversity, or in comparison to other similar species? It may be helpful to contextualize low here or simply rely on the numeric descriptors instead.

The interpretation of Fst values depends on populations of comparison and varies between different species. There is not a good standard to evaluate Fst and nucleotide diversity for the plains bison herds based on the mtDNA control region. We decided to rely on the numeric descriptions as recommended by the Reviewer.

We modified Lines 34-38 from 

“The recent common ancestry of modern bison deriving from small, scattered groups combined with gene flow through foundation and translocation events between herds during the last 100 years, are reflected in low Fst values (0.21), low haplotype (0.48 ± 0.04) and nucleotide (0.004 ± 0.002) diversities, and low mean number of pairwise differences (3.38 ± 1.74).” 

to

“The recent common ancestry of modern bison deriving from small, scattered groups combined with gene flow through foundation and translocation events between herds during the last 100 years, is reflected in Fst value (0.21), haplotype (0.48 ± 0.04) and nucleotide (0.004 ± 0.002) diversities, and mean number of pairwise differences (3.38 ± 1.74).”

We modified Lines 243 – 246 in the original version of the manuscript from

“A low FST genetic distance value of 0.21 (p = 0.000) confirmed low differentiation between the herds. Pairwise FST values between populations ranged from zero to highest FST values observed between Badlands National Park and Grasslands National Park bison (FST = 0.61) and Grasslands National Park and National Elk Refuge bison (FST = 0.61) (Fig 3).” 

to

“A FST genetic distance value of 0.21 (p = 0.000) confirmed some differentiation between the herds. Pairwise FST values between populations ranged from zero to highest FST values observed between Badlands National Park and Grasslands National Park bison (FST = 0.61) and Grasslands National Park and National Elk Refuge bison (FST = 0.61) (Fig 3).” in Lines 246 – 249 in the revised version of the manuscript. 

We modified Lines 320 – 322 in the original manuscript from

“Recent common ancestry of modern bison deriving from small source populations and historic and recent admixture are reflected in low Fst value, low haplotype and nucleotide diversities and low mean number of pairwise differences.”

to

“Recent common ancestry of modern bison deriving from small source populations and historic and recent admixture is reflected in Fst value, haplotype and nucleotide diversities and mean number of pairwise differences.” in Lines 322 – 324 in the revised version of the manuscript.

Page 7, line 137: Is there a citation or other reference available for the specific extraction protocols used?

As stated above in the rebuttal letter, we added a reference [13] for the specific extraction protocol, changing a sentence

“DNA was extracted from 5-6 hair roots or from 3 mm punches of skin tissue using standard Proteinase-K digestion protocols.”

to 

“DNA was extracted from 5-6 hair roots or from 3 mm diameter punches of skin tissue with a standard Proteinase-K digestion protocols used by the Veterinary Genetics Laboratory, U.C. Davies [13].” in Lines 136 – 138 in the revised version of the manuscript.

The full reference [13] was added to References in Lines 605 – 606:

Locke MM, Penedo MC, Bricker SJ, Millon LV, Murray JD. Linkage of the grey coat colour locus to microsatellites on horse chromosome 25. Anim Genet. 2002;33: 329-337.

Page 8, line 166: Perhaps it would also be useful to mention raggedness index here, if Arlequin was also used for this analysis.

Lines 165 – 167 were modified from “Genetic differentiation such as mean pairwise differences, FST, and Analysis of Molecular Variance (AMOVA), Tajima’s D and Fu’ values were calculated using ARLEQUIN version 3.5.2.2 [15].” 

to 

“Genetic differentiation such as mean pairwise differences, FST, and Analysis of Molecular Variance (AMOVA), Tajima’s D, raggedness index, and Fu’ values were calculated using ARLEQUIN version 3.5.2.2 [16].”

Page 8, line 172: It’s slightly unclear to me what is being used as input for the MDS analysis. The authors state that “Genetic distances used for MDS were based on pairwise FST values between populations,” so are the genetic distances simply the matrix of Fst values between populations, or are they being transformed in some way prior to MDS?

The data were not transformed; we used the matrix of Fst values between populations.

Page 12, line 224: Wouldn’t the Fst be the same for any comparison between Grasslands NP and any of the herds where all sampled bison had Haplotype 1, rather than just the two mentioned?

For clarity, the sentence was modified from “Haplotype diversity among bison herds ranged from 0 for herds with only 1 haplotype identified (Badlands National Park, National Elk Refuge, and Theodore Roosevelt National Park) to 0.81 in the Rocky Mountain Arsenal herd where 5 haplotypes were identified.” 

to 

“Haplotype diversity among bison herds ranged from 0 for herds with only 1 haplotype identified to 0.81 in the Rocky Mountain Arsenal herd where 5 haplotypes were identified.” in Lines 232 – 234 in the revised version of the manuscript.

Page 14, line 269: The authors state that MDS reveals only small differences between herds, though I’m not sure this is correct, since MDS as presented isn’t really quantifying the extent of difference but rather showing major axes of variation. Perhaps instead it might be clearer to state something along the lines of “MDS doesn’t reveal strong clustering structure,” if this is what is meant by the authors? However, there does seem to be at least three distinct clusters, as described by the authors.

Line 269 in the original version of the manuscript was changed from “MDS revealed only small differences between populations (Fig 5).” 

to 

“MDS revealed small differences between populations in three clusters (Fig 5).” in Line 272 in the revised version of the manuscript.

Page 16, line 314: Perhaps be “low mean”, instead of “mean low”?

We replaced “mean low” by “low mean” in Line 317 in the revised version of the manuscript. 

Page 17, line 337: The authors suggest that the raggedness index suggests a constant population size throughout the 20th century for the bison metapopulation. However, do we know from the historical record that this isn’t true, with the population increasing throughout the 20th century, particularly at the beginning?

Because the raggedness index was 0.41 with p = 0.70, we cannot reject the null hypothesis of population expansion (Harpending, 1994). 

We removed the sentence “Together with the high raggedness index, these results suggest a fairly constant size of the total combined metapopulation of herds through the twentieth century.” in Lines 337 – 338 in the original version of the manuscript.

Page 25, line 482: The authors mention sampling bias here and later in the discussion. I wonder if it might be possible to show a rarefaction curve of discovered haplotypes with increasing sample numbers or something similar to address how much of the total haplotypic diversity has been sampled and how many new haplotypes would likely be discovered with increased sampling.

Due to the fact that bison are heavily managed in culling strategies and limited in population size based on their location’s available resources, it may not be informative to run a haplotype accumulation analysis. There are too many variables for each population that would not be accounted for in this analysis.

As a final note, it would be great if the authors could indicate where and how the data will be made available (presumably on GenBank?).

We made all data underlying the findings in our manuscript fully available. It has been also stated by both Reviewers in Section 3.

We will submit the mtDNA sequences of haplotypes to GenBank after the publication of this study.

Our own changes not requested by the Editor and Reviewers.

We added Acknowledgements in Lines 568 – 570 in the revised version of the manuscript 

“We thank two Reviewers for providing constructive and valuable comments that helped us to improve the manuscript.”

We changed “Methods” to “Materials and methods” in Line 110 in the revised version of the manuscript according to the PLoS One manuscript body formatting guidelines.

We changed “REFERENCES” to “References” in Line 572 in the revised version of the manuscript according to the PLoS One manuscript body formatting guidelines.

---

## [Decision Letter · Decision Letter 1]

18 Feb 2022

Genetic variation of the mitochondrial DNA control region across plains bison herds in USA and Canada

PONE-D-21-13309R1

Dear Dr. Ovchinnikov,

We’re pleased to inform you that your manuscript has been judged scientifically suitable for publication and will be formally accepted for publication once it meets all outstanding technical requirements.

Kind regards,

Lalit Kumar Sharma

Academic Editor

PLOS ONE

Additional Editor Comments (optional):

Few minor corrections to be incorporated in the final revised manuscript.

1. I saw one small typo in the added reference on line 138, I believe it should be U.C. Davis.

2. Line number 20: Delete (Hap 1 – Hap 11) and replace ‘with’ to ‘and’

Line 98-Were these herds, where both plains and wood bison differentiated?

Also ensure data sequence should be submit at NCBI before publication and accession number need to provide in this MS as these sequences accession number not provided in main text.

Reviewers' comments:

Reviewer's Responses to Questions

**Comments to the Author**

1. If the authors have adequately addressed your comments raised in a previous round of review and you feel that this manuscript is now acceptable for publication, you may indicate that here to bypass the “Comments to the Author” section, enter your conflict of interest statement in the “Confidential to Editor” section, and submit your "Accept" recommendation.

Reviewer #1: All comments have been addressed

Reviewer #3: All comments have been addressed

2. Is the manuscript technically sound, and do the data support the conclusions?

Reviewer #1: Yes

Reviewer #3: Yes

3. Has the statistical analysis been performed appropriately and rigorously? 

Reviewer #1: Yes

Reviewer #3: Yes

4. Have the authors made all data underlying the findings in their manuscript fully available?

Reviewer #1: Yes

Reviewer #3: No

5. Is the manuscript presented in an intelligible fashion and written in standard English?

Reviewer #1: Yes

Reviewer #3: Yes

7. PLOS authors have the option to publish the peer review history of their article (what does this mean?). If published, this will include your full peer review and any attached files.

Reviewer #1: No

Reviewer #3: No

---

## [Editor Report · Acceptance letter]

2 Mar 2022

PONE-D-21-13309R1 

Genetic variation of the mitochondrial DNA control region across plains bison herds in USA and Canada 

Dear Dr. Ovchinnikov:

I'm pleased to inform you that your manuscript has been deemed suitable for publication in PLOS ONE. Congratulations! Your manuscript is now with our production department. 

Kind regards, 

on behalf of

Dr. Lalit Kumar Sharma 

Academic Editor

PLOS ONE